# TIME-SENSITIVE REPLAY FOR CONTINUAL LEARNING

## ABSTRACT

Continual learning closely emulates the process of human learning, which allows a model to learn for a large number of tasks sequentially without forgetting knowledge obtained from the preceding tasks. Replay-based continual learning methods reintroduce examples from previous tasks to mitigate catastrophic forgetting. However, current replay-based methods often unnecessarily reintroduce training examples, leading to inefficiency, and require task information prior to training, which requires preceding knowledge of the training data stream. We propose a novel replay method, Time-Sensitive Replay (TSR), that reduces the number of replayed examples while maintaining accuracy. TSR detects drift in the model's prediction when learning a task and preemptively prevents forgetting events by reintroducing previously encountered examples to the training set. We extend this method to a task-free setting with Task-Free TSR (TF-TSR). In our experiments on benchmark datasets, our approach trains 23% to 25% faster than current task-based continual learning methods and 48% to 58% faster than task-free methods while maintaining accuracy.

## 1 INTRODUCTION

Continual learning aims to emulate the process of human learning and produce a model that learns a sequence of tasks over time. However, these learning systems suffer from *catastrophic forgetting*, whereas humans experience a slow decline in performance (Wang et al., 2020). To overcome this, a strand of continual learning known as replay learning imitates human memory consolidation by reintroducing previously seen data to a model in conjunction with new data (van de Ven et al., 2020; Zha et al., 2020; Wang et al., 2018). As replay generally occurs at every new training batch (Cory et al., 2021; Korycki & Krawczyk, 2021; Wong et al., 2023), a limitation of current research is that replay may occur more than needed, incurring excessive computational costs.

While prior research has predominantly concentrated on the selection of data for replay (Aljundi et al., 2019; Bagus & Gepperth, 2021) and the strategies for reintegrating this data into models (Korycki & Krawczyk, 2021; Wang et al., 2022a; Caccia et al., 2022), there is limited investigation of the temporal aspect of replay mechanisms, specifically, the optimal timing for executing replay operations. We posit that the reiteration of data from tasks that exhibit resistance to memory decay offers limited advantages in terms of enhancing model accuracy. Furthermore, a common assumption in existing work is the availability of task information during the training phase (Cory et al., 2021; Kirkpatrick et al., 2017; Zeng et al., 2019; Klasson et al., 2022). This assumption implies that for a newly encountered example within a data stream, the associated task is explicitly known. Unfortunately, this premise does not align with the reality of many real-world applications, where the partitioning of training data into distinct tasks requires a priori knowledge of the temporal order of examples within the data stream. This inherent limitation renders existing replay-based models impractical for deployment in real-world data stream scenarios (Lee et al., 2020; Ye & Bors, 2022; Aljundi et al., 2019; Jin et al., 2021; Wang et al., 2022b; Rolnick et al., 2019).

To address these challenges, we propose a novel replay strategy called Time Sensitive Replay (TSR). TSR reintroduces data from previously learned tasks only when those tasks are at risk of being forgotten. Specifically, when distribution drift is detected in the model's predictions for a previously learned task, examples from that task are replayed in the subsequent training batch. TSR's primary objective is to reduce the amount of replay required throughout a model's lifespan while maintaining a similar level of accuracy compared to a full-replay model, thereby reducing computational training costs. Additionally, we extend the TSR method to Task-Free Time-Sensitive Replay (TF-TSR),

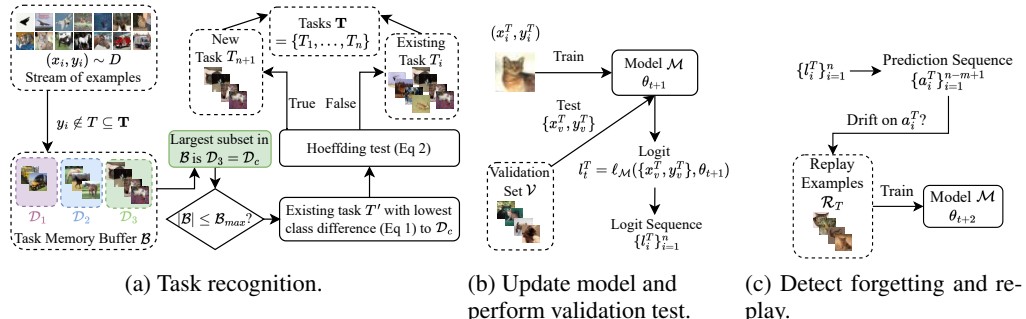

(a) Task recognition.

(b) Update model and perform validation test.

(c) Detect forgetting and replay.

Figure 1: **An overview of the TF-TSR model.** (a) Add a new example to a similar existing task, or create a new task. (b) Update the model on a training example from a known task $T$ and produce a stream of prior logits from the validation set for task $T$. (c) Calculate a moving average from the stream of logits from task $T$ and replay examples from task $T$ if drift exists.

which applies TSR without the need for explicit task information by establishing task boundaries around classes with similar feature loss characteristics. This enables the method to adapt to changing class distributions over data streams without prior knowledge of future tasks. To mitigate the detrimental effects of concurrently replaying dissimilar tasks, which can lead to a more pronounced decrease in model accuracy (De Lange et al., 2023), we propose the use of a class difference metric to estimate class similarity and dynamically group similar classes within tasks.

Figure 1 provides an overview of TF-TSR, divided into three components. The first component, as depicted in Figure 1a, focuses on task recognition. It aims to group similar classes into tasks stored in memory as new examples arrive in the data stream. If an arriving example carries a class not yet associated with any task in memory, it is added to the Task Memory Buffer. This buffer accumulates examples until a sufficiently large sample size is reached for effective task recognition. When the buffer reaches its maximum capacity, denoted as $\mathcal{B}_{max}$, the most frequent class within the buffer becomes a candidate for addition to an existing task in memory or for the creation of a new task. In TF-TSR, the similarity between the most frequent class denoted as $c$, and the tasks in memory are determined using a class difference measure. If the computed similarity is below a certain threshold, a new task is warranted for class $c$, and a new task is created. Specifically, if the average feature loss difference between examples from the task $T'$ and class $c$ exceeds the Hoeffding bound threshold, a new task is created for class $c$. Conversely, if the similarity indicates that class $c$ can be associated with an existing task $T'$, class $c$ is appended to that task. The second component in Figure 1b revolves around model training and validation over previous tasks. When an example from the data stream carries a class already linked to an existing task in memory, the example is utilized for model training and updating internal model parameters. Subsequently, a validation test is conducted on a prior task, referred to as $T$. This test produces a sequence of model predictions represented as logits. The third component, as illustrated in Figure 1c, deals with replaying examples in response to signs of forgetting. A moving average is computed over the sequence of logits obtained during validation on task $T$. If distribution drift is detected within this moving average, stored examples from task $T$ are reintroduced into the training process to update the model. The code can be found at the anonymous link: https://anonymous.4open. science/r/Task-Free-Time-Sensitive-Replay-6F3D/README.md.

## 2 RELATED WORK

This section summarizes existing work regarding replay-based approaches in continual learning. We also provide an overview of task-free continual learning research.

**Replay Learning.** Recently, there has been a growing interest in replay-based research into the selection or construction of replay data and how this data should be reintroduced to the model. Reactive Subspace Buffer (RSB) (Korycki & Krawczyk, 2021) applies concept drift detection in a replay learning context. RSB's replay buffer adapts to drift in a class's incoming data and prioritizes

the replay of examples most representative of the current state of each class. RSB also replays consistently even if a class is not undergoing forgetting. Continual Error Correction (CEC) (Cory et al., 2021) aims to improve existing knowledge and mitigate catastrophic forgetting through replay. CEC is built on a regularization method, Orthogonal Weight Modification (OWM) (Zeng et al., 2019), which reduces the impact of catastrophic forgetting by limiting changes to parameters important to classifying previous tasks. The number of replayed examples increases with the number of tasks, a common problem with replay techniques. Little research has been conducted on when to replay in a continual learning environment. Klasson et al. (2022) uses a Monte Carlo tree search (MCTS) to decide on a replay schedule before model training; however, the schedule cannot change dynamically during training to adapt to changes in the data stream.

**Task-free Learning.** Many methods mentioned in this section heavily rely on the definition of task boundaries; however, tasks are rarely provided in real-world applications. Continual Neural Dirichlet Process Mixture (CN-DPM) (Lee et al., 2020) expands a set of experts, each specializing in a distinct class as new classes are introduced, removing the requirement for task information. CN-DPM introduces short-term memory (STM) to store instances belonging to new classes and to decide when to train a new expert. CN-DPM also measures discrepancies between tasks to ensure that tasks comprise distinct classes. Online Discrepancy Distance Learning (ODDL) (Ye & Bors, 2022) expands on this idea by establishing a theoretical framework to analyze the forgetting behaviour of task-free models. While ODDL and CN-DPM apply STM and task discrepancy to an expert-based model, it may also be useful in establishing task boundaries in a replay-based model.

## 3 NOTATIONS AND PRELIMINARIES

Here, we define the learning settings and notations used to describe our techniques.

**Continual Learning.** Continual learning aims to train a model on a stream of examples. While a typical deep learning method has access to an entire dataset for training, a continual learning method assumes that only an example $(x_t, y_t)$ is available at time $t$ during training. Typically, a continual learning method encounters each example once during training (Chen & Liu, 2018; Jin et al., 2021).

**Task-based Learning.** We define a class as a set of objects with the same classification label $y \in Y$ and a task $T$ as a set of one or more classes (Cory et al., 2021). We assume that examples across all tasks have the same shape and modality. We define an example as an object $(x_t^T, y_t^T)$ belonging to a task $T$. We define a task-free setting as an environment where $T$ is unknown when encountering a new example. TF-TSR creates a set of recognized tasks **T** as more classes are encountered during training. A dataset in a continual learning setting is a sequence of examples $S = \{(x_t^{T_i}, y_t^{T_i})\}_{i=1,t=1}^{\infty}$ such that examples from task $T_i$ do not overlap with examples from task $T_{i+1}$. A continual learning model $\mathcal{M}$ in a task-based environment receives an input example $(x_t^{T_i}, y_t^{T_i}) \in S$ and produces a vector $\hat{y}_t^{T_i} = f_{\mathcal{M}}(x_t^{T_i}, y_t^{T_i}; \theta_t)$ parameterized by some parameter $\theta_t$. The goal is to optimize $\theta_t$ dynamically so that model $\mathcal{M}$ maintains a high prediction accuracy on all observed tasks.

**Confidence.** Confidence is a value ranging between 0 and 1, representing the prior likelihood that a model produced a correct classification. Specifically, for prediction $\hat{y}_t^T = [y_1, \ldots, y_{|Y|}]^{\top}$, where each scalar $y_i$ maps to a corresponding class, confidence $C_t^T = \max(\hat{y}_t^T)$. We define $l_t^T = \ln\left(C_t^T (1 - C_t^T)^{-1}\right)$ as the prediction logit of $x_t^T$. When a stream of $n$ validation examples from task $T$ is input to the model, we denote the resulting sequence of logits as $\{l_i^T\}_{i=1}^n = \{l_1^T, \ldots, l_n^T\}$.

Whenever a new example $(x_{n+1}^T, y_{n+1}^T)$ is encountered in the data stream, $l_{n+1}^T$ is appended to $\{l_i^T\}_{i=1}^n$ resulting in the sequence of logits $\{l_i^T\}_{i=1}^{n+1}$. Logits can occasionally experience radical changes due to anomalous examples, so a moving average of logits is used to smooth the sequence and eliminate irregularities. This moving average, which we refer to as the *prediction sequence*, is defined as $\{a_i^T\}_{i=1}^{n-m+1}$ with window size $m$, where

$$a_i^T = \frac{1}{m} \sum_{j=i}^{i+m-1} l_j^T.$$

We define $m$ as a hyperparameter indicating the sensitivity of the drift detector to individual logits.

**Example.** Consider the MNIST dataset (Deng, 2012) as an example. The MNIST dataset comprises handwritten digits labelled **0** through **9** in 28×28 pixel images, with each image considered an

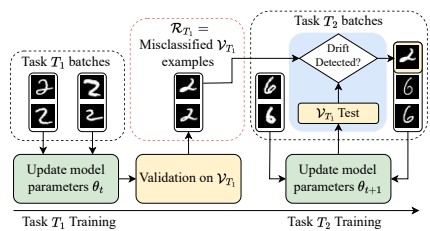

Figure 2: TSR learning the first two tasks in the MNIST dataset.

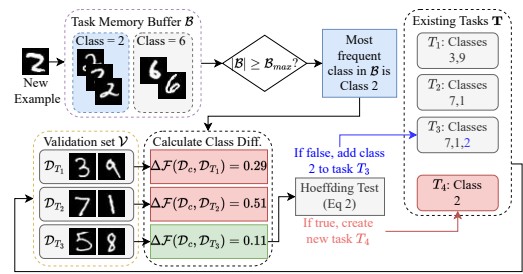

Figure 3: Task identification on the MNIST dataset in TF-TSR.

example. When a model $\mathcal{M}$ takes a vectorized example as input, it produces a logit value for each possible class, in this case, digits **0** through **9**. The prediction is correct if the highest logit corresponds to the correct classification. A task in MNIST may refer to all examples of a particular digit, such as Digit **3**. Thus, $\{l_i^3\}_{i=1}^n$ denotes the sequence of logits corresponding to the prediction for Class **3** for the $n$ most recently classified examples with Label **3**.

## 4 TIME-SENSITIVE REPLAY

This section provides an overview of our Time-Sensitive Replay (TSR) method. Section 4.1 details the architecture of TSR in a task-based environment, and Section 4.2 describes how TSR may be applied to a task-free environment.

### 4.1 TIME-SENSITIVE REPLAY

TSR is a replay-based continual learning technique designed to improve the effectiveness of replay. The goal of TSR is to reintroduce past examples only when a task is experiencing forgetting, thereby enhancing the improvement to model accuracy with each replayed example. An overview of TSR is presented in Figure 2 for a task-based environment. It can be divided into two steps: model update and validation, and forgetting detection and replay. During model update and validation, new examples from the data stream are introduced to the model for training and validation is performed on previous tasks to acquire a prediction sequence. During forgetting detection and replay, the prediction sequence is analyzed for signs of forgetting, and if forgetting is detected, examples from previous tasks are reintroduced to the model for training.

**Model Update and Validation.** Let $(x_t^{T_i}, y_t^{T_i}) \in S$ with $i > 1$ be a newly encountered example from the data stream. First, model $\mathcal{M}$ trains on $(x_t^{T_i}, y_t^{T_i})$ and updates its internal parameters $\theta_t$ to $\theta_{t+1}$. As TSR operates only at the data level, parameter updates are performed by the underlying deep learning architecture. Second, let $\mathcal{V}_{T_i}$ be a validation set with examples belonging to task $T_i$. We sample $n$ examples from $\mathcal{V}_{T_i}$, $\mathcal{V} = \{(x_1^{T_i}, y_1^{T_i}), \ldots, (x_n^{T_i}, y_n^{T_i})\}$, for validation on $\mathcal{M}$. Each example in $\mathcal{V}$ is input to the model and outputs a logit $l_j^{T_i} = \max(f_{\mathcal{M}}(x_j^{T_i}, y_j^{T_i}; \theta_{t+1}))$. We assume that $f_{\mathcal{M}}$ returns logit values directly. Each $l_j^{T_i}$ is appended to the series $\{l_k^{T_i}\}_{k=1}^n$, where the maximum number of logits in the sequence is $n$. When $n$ is exceeded, the oldest logit, which is least relevant to current forgetting trends, is discarded. In practice, the validation process is performed after the model trains on batches of examples.

**Forgetting Detection and Replay.** First, the series of logits $\{l_k^{T_i}\}_{k=1}^n$ is smoothed to reduce the impact of anomalies by calculating a prediction sequence $\{a_k^{T_i}\}_{k=1}^{n-m+1}$ with window size $m$ from $\{l_k^{T_i}\}_{k=1}^n$. To detect forgetting on task $T_i$, TSR monitors the prediction sequence with a drift detector for $T_i$ denoted as $D_{T_i}$. We use Kolmogorov-Smirnov Windowing (KSWIN) (Raab et al., 2020) as the drift detector, and the use of other drift detectors in TSR is explored in Appendix D. Distribution drift detected by $D_{T_i}$ is considered a symptom of forgetting, and replay is performed. TSR, therefore, samples $\mathcal{R} \overset{\mathcal{R}_{max}}{\sim} \mathcal{R}_{T_i}$, where $\mathcal{R}_{T_i}$ is a replay buffer containing misclassified examples from task $T_i$ and $\mathcal{R}_{max}$ is a hyperparameter defining the maximum size for a replay batch. The selected replay

examples $\mathcal{R}$ are introduced to the subsequent training batch upon detection of distribution drift over the prediction sequence for task $T_i$, and reinforce the model's understanding of the task. As an example of the TSR process, let us again consider the MNIST dataset. Figure 2 contains examples of two tasks, $T_1$ has examples of handwritten **2** and $T_2$ has examples of handwritten **6**. First, the model is updated with examples from $T_1$. As $T_1$ is the first task the model learns, there are no previous tasks to replay. Between training batches for $T_2$, suppose the prediction sequence after tests on $\mathcal{V}_{T_1}$ is a decreasing series $\{a_i^{T_1}\}_{i=1}^{10}$. This series is input to the drift detector $D_{T_1}$, which detects distribution drift. A random selection of $\mathcal{R}_{max}$ examples from $\mathcal{R}$ is added to the next training batch for $T_2$, responding to the forgetting event to restore a potential drop in accuracy on $T_1$.

## 4.2 TASK-FREE TIME-SENSITIVE REPLAY

In the previous section, TSR assumes a task $T$ is known for each example. This task information allowed us to initialize drift detectors for each task, establish $\mathcal{V}_T$ for each task, and define task segments for the replay buffer in advance. In reality, task information is often unavailable. To alleviate this, we propose a task-free implementation of TSR, Task-Free Time-Sensitive Replay (TF-TSR), where new tasks are recognized as new classes arrive. Figure 1 shows that TF-TSR has two additional components: a Task Memory Buffer (TMB) and a class difference mechanism. The TMB is a set of examples with classes that do not belong to existing tasks. The class difference mechanism is applied to examples output by the TMB to determine which task the examples belong to. The validation and replay buffers are also altered to sustain the task-free setting. A discussion of the theoretical complexity of TSR and TF-TSR is provided in Appendix C.

**Task Memory Buffer.** Inspired by Short-Term Memory (Lee et al., 2020), we propose the TMB to determine whether a newly encountered class should be incorporated into an existing task or treated as a distinct task. If an example arrives with a class that does not exist within the established task boundaries, that example is passed into the TMB, which we denote as $\mathcal{B}$. Once $|\mathcal{B}| \geq \mathcal{B}_{max}$, where $\mathcal{B}_{max}$ is a hyperparameter, the most frequent class in the TMB is either added to an existing task or a new task is created. A full TMB ensures the model can access a sample of examples from a new class to compare with tasks in memory.

**Class Difference.** When the TMB produces a new class, the model either adds the class to an existing task or creates a new task for the class. To decide which existing task the class should be added to, we match a task containing similar classes to the new class. The similarity of classes within a task is important because training on dissimilar classes can cause a steep decline in model accuracy (De Lange et al., 2023). We therefore propose a class difference measure, denoted $\Delta \mathcal{F}$, to determine the similarity of examples $\mathcal{D}_c$ from class $c$ and examples $\mathcal{D}_T$ from task $T$. Class difference is defined in Equation 1, where $\ell_{\mathcal{M}}$ is the loss function for model $\mathcal{M}$.

$$\Delta \mathcal{F}(\mathcal{D}_c, \mathcal{D}_T) = |\mathbb{E}_{\mathbf{x} \sim \mathcal{D}_c}[\ell_{\mathcal{M}}(\mathbf{x})] - \mathbb{E}_{\mathbf{x} \sim \mathcal{D}_T}[\ell_{\mathcal{M}}(\mathbf{x})]| \tag{1}$$

**Validation and Replay Buffer.** Previously, we have assumed that a validation set is provided and that the replay buffer has access to task information a priori to training. In TF-TSR, we eliminate both of these assumptions. For a set of known tasks $\mathbf{T}$, the replay buffer $\mathcal{R} = \{\mathcal{R}_{T_i} : T_i \in \mathbf{T}\}$ and the validation set $\mathcal{V} = \{\mathcal{V}_{T_i} : T_i \in \mathbf{T}\}$. When a new task is identified, a new subset is initialized in $\mathcal{V}$ and $\mathcal{R}$. A maximum number of examples per subset, $\mathcal{V}_{max}$, is set prior to training. For instance, when $|\mathcal{V}_{T_i}| > \mathcal{V}_{max}$, the oldest example is removed from the set. To initially populate $\mathcal{V}_{T_i}$, all examples from the TMB are placed in $\mathcal{V}_{T_i}$ when a new task is identified. Afterwards, each new example from task $T_i$ has a 10% chance of being added to $\mathcal{V}_{T_i}$. The replay buffer is populated with incorrectly classified examples from each task.

**Task Recognition.** When the TMB is full, the model retrieves a set of examples belonging to the most frequent class $c$, $\mathcal{D}_c$ from the TMB. We then add this class to the task

$$T' = \underset{T \in \mathbf{T}}{\mathrm{argmin}}(\Delta \mathcal{F}(\mathcal{D}_c, \mathcal{D}_T)).$$

A new task is created when $c$ is not similar to any existing task. We use a Hoeffding test with Bonferroni correction (Bifet & Gavalda, 2007) on $\ell_{\mathcal{M}}(\mathcal{D}_c)$ and $\ell_{\mathcal{M}}(\mathcal{D}_{T'})$:

$$\epsilon = \sqrt{|\mathcal{D}_c| \cdot \sigma_{\ell_{\mathcal{M}}}^2 \ln \frac{2|\mathcal{D}_c|}{\delta}} + \frac{4}{3|\mathcal{D}_c|} \ln \frac{2|\mathcal{D}_c|}{\delta} \tag{2}$$

where $\delta$ is the sensitivity to task creation and $\sigma^2_{\ell_{\mathcal{M}}}$ is the variance of $\ell_{\mathcal{M}}$. A new task is created if

$$\left| \overline{\ell_{\mathcal{M}}(\mathcal{D}_c)} - \overline{\ell_{\mathcal{M}}(\mathcal{D}_{T'})} \right| \geq \epsilon.$$

As an example of the TF-TSR task identification process, we consider the MNIST dataset in Figure 3. When an example from a class that does not exist within any known task is encountered, it is added to TMB. When TMB is full, examples with the most frequent class labels are stored while the rest are discarded. In this case, examples with Class 2 are the most common. The set of all examples from the TMB with Class 2 is denoted as $\mathcal{D}_c$. The class difference is then calculated between $\mathcal{D}_c$ and a sample of examples from existing tasks in the validation set $\mathcal{V}$. The existing task $T'$ with the lowest class difference is identified; in the example, this is task $T_3$. A Hoeffding bound test is performed on $\ell_{\mathcal{M}}(\mathcal{D}_c)$ and $\ell_{\mathcal{M}}(\mathcal{D}_{T'})$, and if the inequality is false, Class 2 is added to task $T_3$, otherwise a new task $T_4$ is created for Class 2 and stored in memory.

## 5 EXPERIMENTS

In our experiments, we aim to answer the following research questions. **RQ1:** How can we decrease the number of training examples for a continual learning system? **RQ2:** How can we dynamically detect task boundaries for a continual learning system?

**Datasets.** We use five benchmark datasets and two synthetic dataset generators for our experiments. In the benchmark datasets, we use the MNIST dataset (Deng, 2012) of handwritten digits, Fashion MNIST (Xiao et al., 2017), which contains images of articles of clothing, and CIFAR-100 (Krizhevsky et al., 2009), COIL-100 (Nayar, 1996) and Mini-ImageNet (Vinyals et al., 2016) containing coloured images of various objects and animals. We order these datasets to be *task-sequential*; where tasks are presented in a non-overlapping series one after the other, with each task only appearing once. Further descriptions of the datasets are provided in Appendix E.

In a realistic setting, task boundaries may overlap. To better simulate performance in real-world environments, we create two synthetic dataset generators to simulate environments where tasks intersect, each taking an existing image dataset as an input and producing a different ordering of examples. The first generator selects a proportion $p$ of examples from each task $T_i$ and inserts the chosen examples uniformly throughout every other task $T_j$, where $i, j \in \{1, \ldots, |\mathbf{T}|\}$ and $i \neq j$. This generator is intended to simulate task noise. The second generator selects a proportion $q$ of examples from task $T_i$ and interleaves the chosen examples with the initial examples in $T_{i+1}$, where $i \in \{1, \ldots, |\mathbf{T}| - 1\}$. This generator is intended to simulate a gradual transition between distributions. In our experiments, both $p$ and $q$ range from $0.2$ to $0.8$.

**Baselines.** We compare our results with three task-based baseline models and two task-free baseline models. Our task-based baselines include the regularization model Orthogonal Weights Modification (OWM) (Zeng et al., 2019) and the replay models Continual Error Correction (CEC) (Cory et al., 2021) and Experience Replay (ER) (Riemer et al., 2018). We also define TSR Random, which replays the same number of examples as regular TSR but selects random time intervals to replay for random tasks. We compare these baselines to TSR in a task-based environment. Our task-free baselines include Online Discrepancy Distance Learning (ODDL) (Ye & Bors, 2022), Gradient-based Memory Editing (GMED) (Jin et al., 2021), Continual Learning by Modelling Intra-Class Variation (MOCA) (Yu et al., 2023) and Proxy-based Contrastive Replay (PCR) (Lin et al., 2023). We compare these baselines to TF-TSR in a task-free setting.

**Architectures and Training Details.** We use OWM as the underlying regularisation architecture for TSR and TF-TSR. Task identifiers are only provided for experiments on OWM, CEC and TSR. All results are reported across 10 trials for one epoch, each with different seeds and random task ordering. Each replay-based baseline replays the same number of examples. We randomise the classes in each task for datasets with more than one class per task for task-based experiments. Further details on reproducibility and the hyperparameters for all experiments are provided in Appendix E.1.

**Classification Accuracy *(RQ1, RQ2)*.** Table 1 presents the accuracy of OWM, CEC, TSR, TSR Random and the task-free models TF-TSR, ODDL and GMED over our datasets. Accuracy refers to the average classification accuracy over all tasks. There is no significant difference in accuracy between TSR, TF-TSR and other baseline models for task-based and task-free learning, respectively.

Table 1: Accuracy comparison of task-based and task-free models on a range of benchmark datasets.

| | Methods | MNIST | Fashion MNIST | CIFAR-100 | COIL-100 | Mini ImageNet |
|---|---|---|---|---|---|---|
| **Task-Based** | TSR (Ours) | 88.23±2.43 | 73.54±2.96 | 6.89±1.05 | 8.23±1.11 | 3.52±0.67 |
| | OWM (Zeng et al., 2019) | 80.50±1.92 | 71.02±2.02 | 6.09±0.49 | 7.22±0.67 | 3.19±0.44 |
| | ER (Riemer et al., 2018) | 80.98±0.48 | 72.07±0.43 | 6.14±0.52 | 7.35±0.72 | 3.22±0.46 |
| | CEC (Cory et al., 2021) | 90.45±1.05 | 74.55±1.88 | 7.39±0.50 | 9.05±0.62 | 3.62±0.35 |
| | TSR Random (Ours) | 80.89±2.31 | 72.34±2.12 | 6.15±0.76 | 7.59±1.01 | 3.32±0.32 |
| **Task-Free** | TF-TSR (Ours) | 90.07±1.12 | 75.42±1.62 | 10.45±0.62 | 13.10±3.58 | 3.40±0.48 |
| | ODDL (Ye & Bors, 2022) | 91.31±0.12 | 75.82±0.32 | 7.62±0.54 | 13.06±0.63 | 3.31±0.31 |
| | GMED (Jin et al., 2021) | 88.92±1.40 | 74.85±0.81 | 10.21±1.43 | 13.42±1.51 | 3.65±1.31 |
| | MOCA (Yu et al., 2023) | 90.58±1.52 | 76.01±1.86 | 10.97±1.46 | 13.51±1.72 | 3.86±0.58 |
| | PCR (Lin et al., 2023) | 89.78±0.97 | 75.49±1.02 | 11.03±1.52 | 13.78±1.45 | 4.01±0.62 |

There is no statistically significant difference between our methods and baseline models.

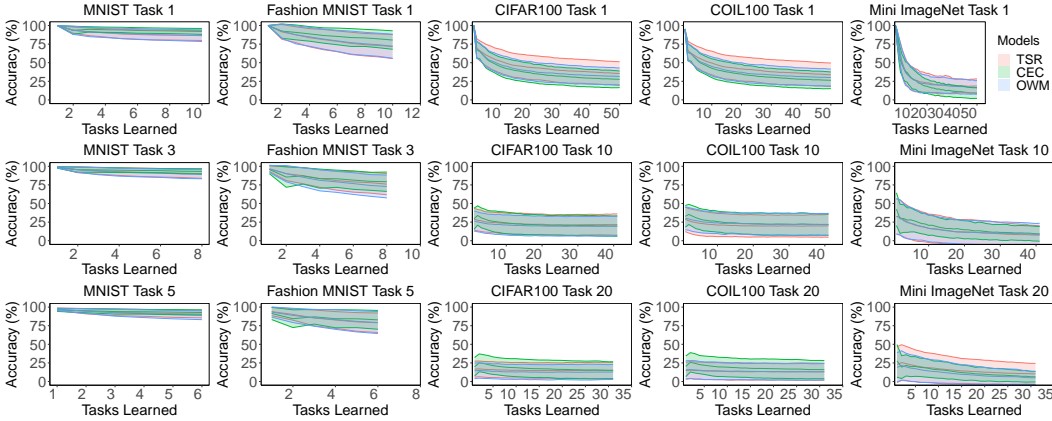

Figure 4: Classification accuracy over a selection of tasks for each model over benchmark datasets.

*Task-Specific Accuracy and Replayed Examples.* We analyse how the models perform over individual tasks during one epoch of model training. As new tasks are introduced, we expect the performance of previous tasks to decrease due to forgetting. Figure 4 compares CEC and TSR accuracy as tasks from benchmark datasets are learned. TSR follows similar forgetting trends to current replay baselines throughout training. Table 2 shows the number of replayed examples across benchmark datasets as the CEC, TSR, and TF-TSR models train. TSR and TF-TSR replay less than current replay baselines.

*Accuracy Per Replay.* Here, we define a metric to formalize the relationship between accuracy and replay. Existing research tends to compare replay-based models by model accuracy alone, however, model accuracy fails to highlight the role that each replayed example plays in improving overall accuracy. We propose a new metric, Accuracy per Replay (APR), which uses the average accuracy improvement from each replayed example in a task to measure the overall effectiveness of replay on that task.

Table 2: Average number of replayed examples.

| Dataset | CEC | TSR | TF-TSR |
|---|---|---|---|
| MNIST | 77.26±23.22 | 5.54±0.65 | **4.21±0.52** |
| Fashion MNIST | 81.67±58.70 | 7.12±0.29 | **5.83±0.24** |
| CIFAR-100 | 225.91±28.54 | 29.64±1.68 | **24.18±2.84** |
| COIL-100 | 255.91±25.97 | **31.08±1.67** | 35.81±2.16 |
| Mini ImageNet | 329.07±74.39 | **172.61±10.53** | 204.43±15.82 |

Table 3: Average APR ($\times 10^{-2}$) across all tasks on benchmark datasets.

| Dataset | CEC | TSR Random | TSR | TF-TSR |
|---|---|---|---|---|
| MNIST | 0.5±0.2 | 0.01±0.1 | **2.1±0.2** | **2.2±0.4** |
| Fashion MNIST | 0.7±0.1 | 0.3±0.1 | **1.6±0.2** | **2.1±0.3** |
| CIFAR-100 | 0.3±0.1 | 0.1±0.2 | **0.6±0.1** | **0.8±0.1** |
| COIL-100 | 0.5±0.1 | 0.1±0.1 | **1.1±0.2** | **1.2±0.1** |
| Mini ImageNet | 0.1±0.1 | 0.0±0.1 | 0.3±0.1 | **0.4±0.1** |

CEC, TSR and TF-TSR models use the replay mechanisms from OWM as a base model. If a model does not replay, it performs identically to OWM if trained under the same conditions. Moreover,

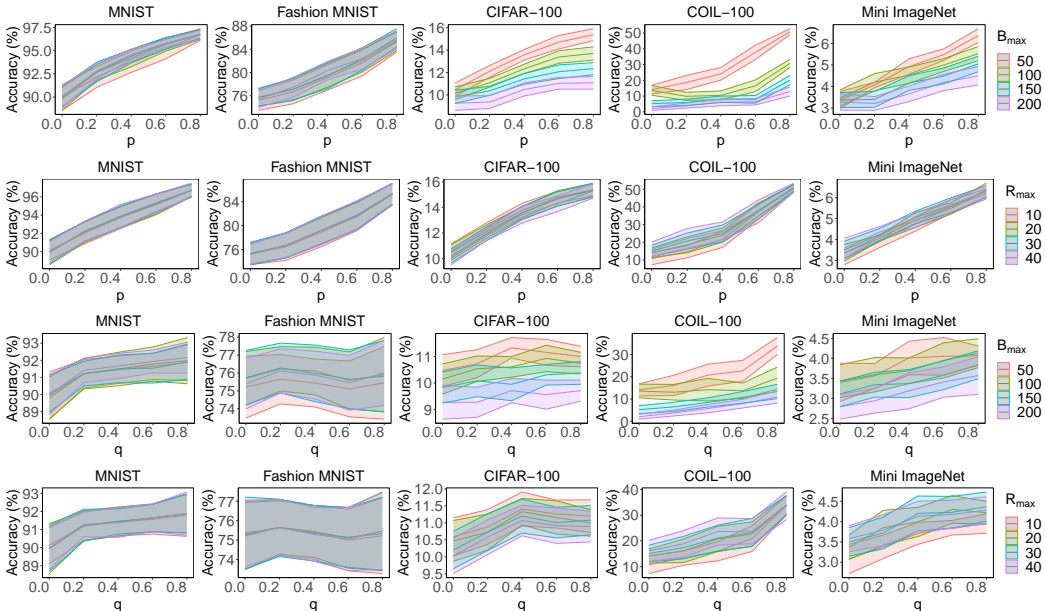

Figure 5: Accuracy comparison of TF-TSR on synthetic image datasets with $p$ and $q$ ranging from 0.0 to 0.8 with varied $\mathcal{B}_{max}$ and $\mathcal{R}_{max}$. Each row varies a single hyperparameter.

any increased accuracy observed over the lifespan of CEC, TSR or TF-TSR compared to OWM is due to the replayed data introduced to the training data.

Equation 3 defines APR for task $T$ in model $\mathcal{M}$ ($APR_{\mathcal{M}}^T$), where $acc_{\mathcal{M}}^T$ is the classification accuracy of a model $\mathcal{M}$ with replay over task $T$, $acc_{\mathcal{M}'}^T$ is the classification accuracy of the same model without replay, $\mathcal{M}'$, over task $T$, and $R_{\mathcal{M}}^T$ is the number of examples from $T$ replayed by $\mathcal{M}$. In our case, $\mathcal{M}$ is CEC, TSR, TF-TSR or TSR Random, and $\mathcal{M}'$ is OWM. The constant $\gamma$ is a small positive for Laplace correction. Here, we set $\gamma = 0.001$.

$$APR_{\mathcal{M}}^T = \frac{acc_{\mathcal{M}}^T - acc_{\mathcal{M}'}^T}{R_{\mathcal{M}}^T + \gamma} \tag{3}$$

Table 3 shows the average $APR_{\mathcal{M}}^T$ over all tasks. OWM is $\mathcal{M}'$ for every model, and $\mathcal{M}$ is CEC, TSR, TF-TSR or TSR Random. We observe that TSR produces a higher average $APR_{\text{TSR}}^T$ than $APR_{\text{CEC}}^T$, indicating that replayed examples after a forgetting event is detected have increased influence on model accuracy. On average, $APR_{\text{TFTSR}}^T$ is highest. This implies that drift detection for forgetting performs more accurately over classes with a similar class difference.

**Sensitivity Tests *(RQ2)*.** We conduct sensitivity tests for two key hyperparameters in TF-TSR: the size of the replay batch $\mathcal{R}_{max}$, and the maximum size of the TMB $\mathcal{B}_{max}$. By varying these hyperparameters and observing the corresponding effects on the model's accuracy, we aim to investigate their influence and identify the optimal configurations. Figure 5 provides the accuracy of TF-TSR on synthetic datasets for a range of $p$ and $q$ when $\mathcal{B}_{max}$ and $\mathcal{R}_{max}$ are varied. Further sensitivity experiments are provided in Appendix E.4.

**Computational Performance *(RQ1)*.** We present the time requirements of TF-TSR and TSR compared to CEC and OWM. The advantage of TSR is that it replays fewer examples than standard replay methods; thus, we expect that this reduces training time. Table 4 shows the time to train each model on benchmark datasets. Timings are averaged over all trials in an identical environment. OWM does not replay any examples and thus has the lowest training time. TSR's training time is lower than CEC, with a statistically significant difference over larger datasets such as CIFAR-100, COIL-100 and Mini ImageNet. TF-TSR is more computationally expensive than TSR as it must recognize new tasks; however, TF-TSR is between $48\%$ to $58\%$ faster than benchmark task-free learning methods on large image datasets.

Table 4: Average training time in seconds on the benchmark image datasets.

|  | Methods | MNIST | Fashion MNIST | CIFAR-100 | COIL-100 | Mini ImageNet |
|---|---|---|---|---|---|---|
| Task-Based | **TSR (Ours)** | **6.12±1.57** | **6.74±1.23** | **43.49±5.58** | **30.72±6.09** | **89.07±18.40** |
|  | OWM (Zeng et al., 2019)* | 2.81±0.89 | 2.84±0.93 | 14.40±0.93 | 14.34±0.95 | 28.61±1.38 |
|  | ER (Riemer et al., 2018) | 7.42±1.01 | 8.22±1.13 | 52.32±3.51 | 39.92±3.12 | 104.23±7.63 |
|  | CEC (Cory et al., 2021) | **7.65±1.12** | **8.43±1.23** | 56.54±4.65 | 43.01±4.12 | 118.76±9.80 |
| Task-Free | TF-TSR (Ours) | **11.25±1.57** | **12.70±0.93** | **126.66±38.00** | **131.36±36.62** | **199.63±47.11** |
|  | ODDL (Ye & Bors, 2022) | 14.63±1.05 | 16.38±0.78 | 506.64±9.50 | 442.26±10.46 | 972.15±18.84 |
|  | GMED (Jin et al., 2021) | **10.22±1.30** | **14.04±0.72** | 241.32±12.67 | 252.36±12.21 | 481.08±23.55 |
|  | MOCA (Yu et al., 2023) | 30.31±1.19 | 32.95±2.28 | 512.71±26.35 | 489.12±24.67 | 1078.18±63.19 |
|  | PCR (Lin et al., 2023) | 20.63±1.36 | 23.49±2.76 | 462.63±21.46 | 398.82±34.18 | 786.51±68.37 |

\* OWM does not replay and is the baseline model for CEC and TSR.

**Ablation Experiments** *(RQ1, RQ2).* We conduct ablation experiments to assess the significance of removing the replay and task recognition components from TF-TSR. Table 5 shows results for two ablation experiments on TF-TSR over synthetic image datasets. The first experiment, replay ablation, removes TF-TSR's ability to revisit past data by eliminating the memory replay mechanism while maintaining the task recognition module. The model is trained on a sequence of tasks without the ability to re-experience previously encountered data. In the second experiment, task ablation, we remove TF-TSR's task recognition module and maintain the replay module. Instead, each new class encountered is assigned an individual task.

Table 5: Accuracy of TF-TSR ablation studies on synthetic datasets.

| Ablation Method | Dataset | $p$ | | | | |
|---|---|---|---|---|---|---|
|  |  | 0 | 0.2 | 0.4 | 0.6 | 0.8 |
| Replay Ablation | MNIST | 89.83±1.29 | 91.79±1.13 | 93.68±1.20 | 95.15±1.12 | 96.77±0.71 |
|  | Fashion MNIST | 75.10±1.57 | 75.52±2.79 | 78.76±2.49 | 81.39±2.39 | 85.29±1.77 |
|  | CIFAR-100 | 8.29±0.80 | 12.38±0.57 | 13.94±0.41 | 14.89±0.49 | 15.50±0.44 |
|  | COIL-100 | 8.83±4.63 | 8.74±4.02 | 17.43±4.69 | 34.67±5.72 | 50.19±2.16 |
|  | Mini ImageNet | 2.57±0.32 | 3.71±0.36 | 4.57±0.32 | 5.48±0.32 | 6.17±0.32 |
| Task Ablation | MNIST | 89.89±1.21 | 91.94±1.07 | 93.66±1.21 | 95.16±1.15 | 96.73±0.75 |
|  | Fashion MNIST | 74.97±1.80 | 75.59±2.74 | 76.77±2.49 | 81.48±2.36 | 85.22±1.77 |
|  | CIFAR-100 | 9.12±0.80 | 10.35±0.56 | 12.26±0.58 | 14.82±0.53 | 15.52±0.48 |
|  | COIL-100 | 7.33±3.67 | 10.98±3.84 | 18.19±4.73 | 36.47±3.69 | 50.92±2.28 |
|  | Mini ImageNet | 2.65±0.31 | 3.73±0.27 | 4.58±0.32 | 5.49±0.23 | 6.22±0.27 |

| Ablation Method | Dataset | $q$ | | | | |
|---|---|---|---|---|---|---|
|  |  | 0 | 0.2 | 0.4 | 0.6 | 0.8 |
| Replay Ablation | MNIST | 89.83±1.29 | 91.24±0.84 | 91.35±0.75 | 91.67±0.85 | 91.82±1.29 |
|  | Fashion MNIST | 75.10±1.57 | 75.62±1.44 | 75.30±1.47 | 74.92±1.49 | 75.28±2.01 |
|  | CIFAR-100 | 8.29±0.80 | 10.68±0.61 | 11.45±0.52 | 11.46±0.48 | 11.42±0.39 |
|  | COIL-100 | 8.83±4.63 | 6.99±3.43 | 10.71±4.49 | 12.17±4.47 | 28.24±5.04 |
|  | Mini ImageNet | 2.57±0.32 | 2.98±0.33 | 3.26±0.31 | 3.39±0.24 | 3.58±0.23 |
| Task Ablation | MNIST | 89.89±1.21 | 91.30±0.79 | 91.33±0.74 | 91.64±0.89 | 91.79±1.21 |
|  | Fashion MNIST | 74.97±1.80 | 75.60±1.38 | 75.23±1.36 | 74.81±1.54 | 75.16±2.09 |
|  | CIFAR-100 | 9.12±0.80 | 10.80±0.67 | 11.47±0.56 | 11.42±0.48 | 11.29±0.46 |
|  | COIL-100 | 7.33±3.67 | 7.22±3.84 | 11.63±4.86 | 13.16±4.90 | 27.95±5.61 |
|  | Mini ImageNet | 2.65±0.31 | 2.99±0.30 | 3.38±0.32 | 4.31±0.35 | 4.82±0.37 |

## 6 CONCLUSION

We proposed a selective time-based replay approach that increases the effectiveness of each replayed example and thereby decreases the computational cost of a replay model. We further posit that the need for task information for replay methods can be avoided by dynamic task recognition during training. We developed the Time Sensitive Replay (TSR) method, which improves training time requirements by implementing replay only when forgetting is detected on previously learned tasks. Beyond that, we extended the TSR method to Task-Free Time-Sensitive Replay (TF-TSR), which eliminates the need for explicit task information. We demonstrated that our models decrease the number of training replay examples, thereby decreasing training time while achieving similar accuracy compared to current state-of-the-art task-free replay methods.

## 7 ETHIC STATEMENT

No human subjects were involved during the research and development of this work. Our experiments were conducted on publicly available benchmark datasets in a controlled environment. Thus, our work has minimal ethical concerns.

## 8 REPRODUCIBILITY STATEMENT

We conduct all experiments ten times with randomized task order to avoid bias to a particular dataset ordering and ten times for each task order to avoid bias due to a random seed. Each experiment is, therefore, performed one hundred times. Our results report the mean and standard deviation values to avoid bias from dataset ordering and random seeds. In Appendix E.1, we provide the full details of our experimental settings.

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

## A   APPENDIX

In this appendix, we provide the pseudocode implementation of the TSR and TF-TSR algorithms, we present an analysis of the theoretical complexity of the TSR and TF-TSR models, we discuss the use of drift detectors in TSR and describe a range of different drift detection methods, we give a detailed description of the datasets and hyperparameters adopted in our experiments, and finally provide additional experimental results. In particular, we provide additional results of sensitivity tests and comparisons between drift detectors, which validate our motivation and contribution.

## B   PSEUDOCODE FOR TSR AND TF-TSR

We present the pseudocode for the TSR and TF-TSR methods described in Sections 4.1 and 4.2 respectively. Algorithm 1 provides an overview of the TSR process and assumes a task-based environment. We leave the drift detection method $D_{T_i}$ arbitrary in this pseudocode as drift detection methods are not the focus of this work, however unless otherwise stated, KSWIN is used in our experiments. For simplicity, we use $\ell_{\mathcal{M}}(\mathcal{V})$ to denote the set of model loss values corresponding to a set of examples $\mathcal{V}$.

---

**Algorithm 1** Time-Sensitive Replay (TSR)

---

**Input:** $S$ - Training dataset, $m$ - Window for moving average, $\mathcal{V}_T$ - Validation set for task $T$, $\alpha$ - Batch Size, $n$ - Validation batch size, $\mathcal{R}_{max}$ - Maximum replayed examples per batch.
**Output:** Model $\mathcal{M}$ trained on all tasks

  Initialize Model $\mathcal{M}$
  $\mathbf{T} \sim S$                                                     $\triangleright$ Set of tasks split from the dataset
  **for** $T_i \in \mathbf{T}$ **do**
      $\mathcal{R}_{T_i} \leftarrow \emptyset$
      $D_{T_i} \leftarrow \text{DriftDetector}()$
      $B_{T_i} \overset{\alpha}{\sim} T_i$                            $\triangleright$ Sample a batch of data with size $\alpha$ from $T_i$
      **for** Batch $B \in B_{T_i}$ **do**
          **for** $T_v \in \{T_1, \ldots, T_{i-1}\}$ **do**
             $\mathcal{V} \overset{n}{\sim} \mathcal{V}_{T_v}$               $\triangleright$ Sample a batch of size $n$ from $V_{T_v}$
             $\{l_i^{T_v}\}_{i=1}^n \leftarrow \ell_{\mathcal{M}}(\mathcal{V})$               $\triangleright$ Model prediction
             $\{a_i^{T_v}\}_{i=1}^{n-m+1} \leftarrow \text{avg}(\{l_i^T\}_{i=1}^n)$
             **if** $D_{T_v}(\{a_i^{T_v}\}_{i=1}^{n-m+1})$ **then**         $\triangleright$ Drift detected
                $B \leftarrow B \cup \mathcal{R}_{T_v}$
             **end if**
          **end for**
          Train $\mathcal{M}$ on $B$ with regularization
      **end for**
      **for** $\mathcal{V}_T \in \{\mathcal{V}_{T_1}, \ldots, \mathcal{V}_{T_i}\}$ **do**
         $\mathcal{R}_T^{(\text{all})} \leftarrow$ Misclassified examples from $\mathcal{V}_T$
         $\mathcal{R}_T \overset{\mathcal{R}_{max}}{\sim} \mathcal{R}_T^{(\text{all})}$
      **end for**
  **end for**

---

Algorithm 2 provides an overview of the TF-TSR method. The primary difference to TSR is the need to recognize tasks dynamically as examples arrive, and the replay buffer $\mathcal{R}$ is populated according to known tasks. For simplicity, we use the function $\text{TSR}(\mathbf{T}, \mathcal{R})$ to denote the model training and replay process with detected tasks $\mathbf{T}$ and the current replay buffer $\mathcal{R}$.

In Section E.1, we provide a link to the implementation of Algorithms 1 and 2 in Python 3 and TensorFlow, accompanied by example commands with preset hyperparameters. Instructions for running the code are included in the repository as well as a list of dependencies.

---

**Algorithm 2** Task-Free Time-Sensitive Replay (TF-TSR)

---

**Input:** $S$ - Training dataset, $\mathcal{B}_{max}$ - Maximum capacity for TMB, $\delta$ - Task creation sensitivity.
**Output:** Model $\mathcal{M}$ trained on all tasks

   $\mathbf{T} \leftarrow \emptyset$
   $\mathcal{B} \leftarrow \emptyset$                                             ▷ Task Memory Buffer.
   $\mathcal{R} \leftarrow \emptyset$                                               ▷ Replay Buffer.
   **for** $(x, y) \in S$ **do**
      **if** $y \notin \mathbf{T}$ **then**
         $\mathcal{B} \leftarrow \mathcal{B} \cup (x, y)$
      **else**
         **for** $(x_i, y_i) \in \mathcal{R}$ **do**                     ▷ Remove oldest example with label $y$.
            **if** $y == y_i$ **then**
               $x_{\text{old}} \leftarrow x_i$
               break
            **end if**
         **end for**
         $\mathcal{R} \leftarrow \mathcal{R} \setminus (x_{\text{old}}, y)$
         $\mathcal{R} \leftarrow \mathcal{R} \cup \{(x, y)\}$
      **end if**
      **if** $|\mathcal{B}| \geq \mathcal{B}_{max}$ **then**                        ▷ Add $y$ to existing task.
         $\mathcal{D}_c \leftarrow \{(x_i, y_i) \in \mathcal{B} : y_i = \underset{y_j}{\text{argmax}} |\{(x_j, y_j) \in \mathcal{B}\}|\}$
         **for** $T \in \mathbf{T}$ **do**
            $\mathcal{D}_T \leftarrow \{(x_i, y_i) : (x_i, y_i) \in \mathcal{R}, y_i \in T\}$
         **end for**
         $T' \leftarrow \underset{T \in \mathbf{T}}{\text{argmin}}(\Delta \mathcal{F}(\mathcal{D}_c, \mathcal{D}_T))$
         $d \leftarrow \ln(2|\mathcal{D}_c|/\delta)$
         $\epsilon \leftarrow \sqrt{|\mathcal{D}_c| \cdot \sigma^2_{\ell_{\mathcal{M}}} d + (4d)/(3|\mathcal{D}_c|)}$
         **if** $\left| \overline{\ell_{\mathcal{M}}(\mathcal{D}_c)} - \overline{\ell_{\mathcal{M}}(\mathcal{D}_{T'})} \right| \geq \epsilon$ **then**
            $\mathbf{T} \leftarrow \mathbf{T} \cup \{y\}$                       ▷ Add new task.
         **else**
            $\mathbf{T} \leftarrow \mathbf{T} \setminus T'$                    ▷ Add $y$ to existing task.
            $\mathbf{T} \leftarrow \mathbf{T} \cup \{T' \cup y\}$
         **end if**
      **end if**
      $\text{TSR}(\mathbf{T}, \mathcal{R})$                             ▷ Perform TSR on $T \in \mathbf{T}$.
   **end for**

---

## C   Theoretical Complexity

This section discusses the computational complexity of the TSR and TF-TSR methods.

**Replay Complexity.** The computational complexity of TSR is dependent on the number of replayed examples. In the best case, forgetting is not detected on any task, and TSR does not replay. In this case, the number of additional training examples $E$ is 0, and the computational complexity is identical to the base model. We use OWM (Zeng et al., 2019) as the base model in our experiments. OWM has complexity $O(N_n N_w^2)$, where $N_n$ is the total number of neurons and $N_w$ is the number of input weights per neuron. In the worst case, a replay is performed on every previous task in every batch. This is similar to the estimation used by Cory et al. (2021) as the same policy of selecting misclassified examples from previous tasks is used. We assume that $0 < \bar{x}_i < 1$ and predict $E$ according to Equation 4, where $\bar{x}_i$ is the expected mean accuracy over task number $i$, $v_i$ is the number of validation examples for task number $i$, and $F_i$ is the expected number of forgetting events detected on task number $i$ over the lifespan of the model. TF-TSR has additional computational costs from the process of task boundary identification. The computational cost of determining which task a new class is added to is $O(2|\mathcal{D}_c||\boldsymbol{T}|)$.

$$E = \sum_{i=1}^{|\boldsymbol{T}|} F_i \cdot (|\boldsymbol{T}| - i)(1 - \bar{x}_i)v_i. \tag{4}$$

We use Equation 4 to inform a validation set size $\mathcal{V}_{max}$. In task-based settings, we acquire $\mathcal{V}$ by splitting the training dataset, so any increase in the size of $\mathcal{V}$ decreases the size of the training set.

We compare Equation 4 with the expected number of additional training samples for a full replay model, $E_R$ which is given in Equation 5, where $B$ is the number of trained batches.

$$E_R = \sum_{i=1}^{|\boldsymbol{T}|} \frac{|T_i|}{\alpha}(|\boldsymbol{T}| - i)(1 - \bar{x}_i)v_i. \tag{5}$$

TSR is more computationally efficient compared to baseline replay models when $E_R - E$ is large. This will be true when $F_i << \frac{|T_i|}{\alpha}$.

**Validation Complexity.** More validation examples are tested when TSR has trained on more tasks, so the computational cost of the method increases with the number of trained tasks. With the size of each task used for training $|T_i|$ and the batch size $\alpha$, the total number of validation examples tested after $x$ tasks is $V_{\text{Test}} = n\sum_{i=1}^{x} \frac{i|T_i|}{\alpha}$.

To constrain computational costs as the number of tasks increases, we set a hyperparameter $n$ as the sample size for $\mathcal{V}$ tested after each batch. We demonstrate in our experiments that TSR is effective even with a low $n$. We therefore need to perform $V_{\text{Test}}$ forward-passes over our neural network. For simplicity, we assume that a simple artificial neural network with $dim(x)$ inputs, $|Y|$ outputs, and $H$ hidden layers with $h_i$ neurons on layer $i$. We provide the complexity of validation tests in Equation 6.

$$\Theta\left(V_{\text{Test}}\left(dim(x)h_1 + h_H|Y| + \sum_{i=1}^{H-1} h_i h_{i+1}\right)\right) \tag{6}$$

## D    DISCUSSIONS ON DRIFT DETECTORS

This section describes the motivation for using drift detectors on output model logits to detect forgetting and compares two different drift detector candidates for the TSR method. Particularly, we compare the Drift Detection Method (DDM) and Kolmogorov-Smirnov Windowing (KSWIN) and justify our decision in Section 4.1 to use KSWIN in our experiments.

Figure 6 shows the logit score and confidence of a single classifier in the MNIST dataset trained on OWM (Zeng et al., 2019) as that task experiences forgetting. Logits change more immediately in response to a forgetting event, so we monitor logits for drift during training.

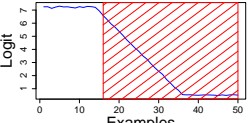 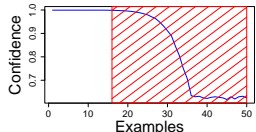 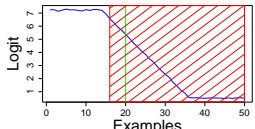

(a) An exert of logit values for an MNIST task in OWM.

(b) An exert of confidence values for an MNIST task in OWM.

(c) Logit values from (a) with a line representing the point at which DDM detects a change in distribution.

Figure 6: Logit and confidence values for a stream of examples during training on a single MNIST task. Examples within the red region were misclassified by the model.

### D.1    DRIFT DETECTION METHOD

Drift Detection Method (DDM) (Gama et al., 2004) is a method of drift detection based on Probably Approximately Correct (PAC) learning that for a stationary data distribution, the model's error rate decreases as the number of analysed examples increase. For a replay model, we expect that error will decrease as the model trains on more data. After the model trains on a new example, DDM calculates whether the error rate exceeds a given threshold.

$$p_t + s_t \leq p_{min} + \lambda s_{min} \tag{7}$$

If the inequality in Equation 7 is true, where $\lambda$ is the hyperparameter defining sensitivity, $p_t$ and $s_t$ are the error rate and standard deviation respectively at the current instant of time $t$ and $p_{min}$ and $s_{min}$ are the minimum error rate and the minimum standard deviation, DDM will return that a change has been detected. We apply DDM to a section of logits produced by a model training on the MNIST dataset as a task experiences a forgetting event. For this example, we set $\lambda = 3.0$.

Figure 6 shows that DDM is successful in quickly identifying a decrease in logit values in this task. As DDM was designed for multi-class imbalanced data streams with different drift types, which is the scenario in a task-based continual learning setting, DDM is an appropriate candidate for TSR.

### D.2 Kolmogorov-Smirnov Windowing

Kolmogorov-Smirnov Windowing (KSWIN) (Raab et al., 2020) is a concept drift detection method predicated upon the Kolmogorov-Smirnov (KS) test. This test is capable of identifying changes in data or performance distributions without any underlying assumptions regarding their specific distribution. It is therefore a valid candidate to detect drift in a stream of logit predictions.

KSWIN maintains a sliding window $\Phi$ of size $n$, where the most recent $r$ examples of $\Phi$ are assumed to represent the most recent concept, $R$. From the first $n - r$ examples of $\Phi$, $r$ examples are selected uniformly, thereby approximating the previous window, $W$.

The KS-test is conducted on the windows $R$ and $W$, which are of the same size. This test compares the distance between the empirical cumulative data distribution $dist(R, W)$.

$$dist(R, W) > \sqrt{-\frac{ln(\alpha)}{r}} \tag{8}$$

KSWIN detects drift if Equation 8 is true, where $\alpha$ is the probability for the test statistic of the KS-test.

## E Experimental Setup and Results

In this section, we first outline steps to reproduce our experiments in Section 5 and provide a link to a code repository implementing TF-TSR. We then describe the benchmark image datasets used in our experiments and their preprocessing steps. We also show TSR results when different drift detectors are used, as well as when parameters specific to TSR are varied. Finally, we present complete results for TF-TSR sensitivity experiments expanding on the content in Section 5. We use parameters identical to those used for experiments in the core material.

### E.1 Reproducability

We perform 10 trials for each experiment, composing tasks of random classifiers and randomizing the task ordering for each trial. This ordering is determined before each trial so that each model is trained on the same number of examples from the same tasks. Each trial for each method is also trained on identical examples. Each trial is repeated 10 times with random examples. Training and testing sets are randomized between examples. This results in 100 experiments for each set of experiments. The training and testing set sizes are shown in Appendix 6. All experiments are performed on one NVIDIA Tesla V100 GPU. Our code is available at `https://anonymous.4open.science/r/Task-Free-Time-Sensitive-Replay-6F3D/README.md`. A continual learning system's results are susceptible to the ordering of tasks, as previous research has indicated that the shaping of data can influence how such models perform (Krueger & Dayan, 2009). We, therefore, vary the ordering and composition of tasks for our experiments. For MNIST and Fashion MNIST, each task contains only one class. For CIFAR-100, COIL-100, and Mini-ImageNet, each task contains two classes.

### E.1.1 HYPERPARAMETERS

Table 6 outlines the hyperparameters we selected for our experiment. The hyperparameters $\delta_{\mathrm{OWM}}$ and $\lambda$ are directly associated with the original OWM method (Zeng et al., 2019), whereas the hyperparameter $\kappa$ corresponds to the model learning rate. Values for $\kappa$ were acquired through a grid search. The batch sizes $\alpha$ and number of layers are similar to settings from CEC (Cory et al., 2021). Hyperparameter $\alpha_D$ is the error threshold of the drift detection method used in TSR and is not applicable to models that do not utilize TSR. We set $\alpha_D = 0.01$ for all experiments. As we have employed KSWIN in our experiments, the value of $\alpha_D$ is not recommended for other drift detectors. The parameters for TSR are $n = 50$ and $m = 2$ for all experiments. The parameters for TF-TSR are $\delta = 0.8$, $\mathcal{R}_{max} = 20$, $\mathcal{B}_{max} = 50$ and $\mathcal{V}_{max} = 50$ unless otherwise stated.

Table 6: Hyperparameters used during experimentation.

| Dataset | $\delta_{\mathrm{OWM}}$ | $\lambda\,(10^{-3})$ | $\kappa$ | $\alpha$ | Layers |
|---|---|---|---|---|---|
| MNIST | 0.6 | 1.00 | 0.001 | 40 | 3 |
| Fashion MNIST | 0.6 | 1.00 | 0.020 | 40 | 3 |
| CIFAR-100 | 1.0 | 0.01 | 0.020 | 64 | 2 |
| COIL-100 | 1.0 | 0.01 | 0.020 | 64 | 2 |
| Mini ImageNet | 1.0 | 0.01 | 0.020 | 64 | 2 |

Models are trained over one epoch to mirror realistic continual learning environments where examples are rarely repeated in the data stream. Table 7 shows the training and testing set sizes.

Table 7: Dataset splits used during experimentation.

| Dataset | Train Size | Val Size* | Test Size | Tasks |
|---|---|---|---|---|
| MNIST | 55000 | 5000 | 10000 | 10 |
| Fashion MNIST | 55000 | 5000 | 10000 | 10 |
| CIFAR-100 | 55000 | 5000 | 10000 | 50 |
| COIL-100 | 5500 | 500 | 1200 | 50 |
| Mini ImageNet | 45000 | 5000 | 10000 | 50 |

* Only set for task-based models.

### E.1.2 DATASETS

**MNIST.** The MNIST dataset (Deng, 2012) consists of $60,000$ training images and $10,000$ test images of 28x28 pixel greyscale images of handwritten digits 0-9. Our validation set is a random sample of $10\%$ of the training images, resulting in $55,000$ training instances and $5000$ validation instances. Each set is composed of ten individual classes which we do not split further into tasks. For our training set, in order to simulate a stream of data under continual learning conditions, we randomly select $5\%$ of training images to produce approximately 250 images for each of the ten classes in each sample. For our validation set, we have 5000 examples and 500 images for each task. Our test set of size $10,000$ contains approximately 1000 instances of each of the 10 classes and is used to evaluate model accuracy.

**Fashion MNIST.** The Fashion MNIST dataset (Xiao et al., 2017) is similar to the MNIST dataset, only it consists of 28x28 grayscale images of various articles of clothing. A series of 10 classes map each image to a type of clothing such as dress, bag, coat or ankle boot. Fashion MNIST shares the same dataset size and the same training and testing splits as the original MNIST, and therefore we use the same validation split.

**CIFAR-100.** The CIFAR-100 dataset (Krizhevsky et al., 2009) consists of $60,000$ 32x32 colour images and 100 classes representing an animal or mode of transportation. There are 600 images per class. The training set consists of $50,000$ images with 500 images per class, and the testing set consists of $10,000$ images with 100 images per class. Since the size of the dataset is identical to MNIST, we use the same split for our validation set. The validation set contains 5000 images with 50 images per class. Each image is photo-realistic containing only one class to which it is relevant. The object in the image may be seen from a variety of viewpoints and may be partially obstructed. The CIFAR-100 dataset is a more complex dataset with more classes than the MNIST datasets, and so is a further challenge for TSR. As CIFAR-100 contains 100 classes, each task contains two individual classes. This results in 50 tasks where each set of two classes is randomly selected for each sample.

**COIL-100.** The COIL-100 (Nayar, 1996) dataset contains 7200 128x128 images of 100 classes each representing a household object. There are 72 images per object each posed at a different angle. Our training set consists of 60 examples for each class resulting in a training set of size 6000 and a test set of size 1200. The validation set is split from the training set with the size of 500.

**Mini ImageNet.** The Mini ImageNet dataset Ravi & Larochelle (2016) contains 60000 coloured images scaled to 64x64 evenly spread over 100 classes. Classes consist of everyday items, vehicles and animals. Our training set consists of 50000 images and our test set consists of 10000 images. Each class in our training set has 500 corresponding examples. The validation set is split from the training set with a size of 5000.

**Synthetic Datasets.** We alter the ordering of examples in the above datasets to create synthetic datasets. Our synthetic datasets contain a different distribution of tasks and ensure that task boundaries are not distinct, thereby testing the task identification mechanism and the forgetting detection mechanism in realistic scenarios.

*Distribution 1.* The first generator simulates noise throughout all tasks. First, tasks are ordered sequentially as described in previous datasets. A proportion $p$ of examples from each task $T_i$ is selected, extracted from the dataset, and randomly inserted into all other tasks $T_j$, where $i \neq j$. This generator challenges the replay and task recognition mechanisms as examples belonging to any task may be encountered at any time during training.

*Distribution 2.* The second generator creates a gradual shift between distributions, where previously tasks would abruptly change during training. First, tasks are ordered sequentially as before. Then, for each task $T_i$, a proportion $q$ of examples are randomly selected and extracted from the dataset. These examples are then interleaved with examples form task $T_{i+1}$ until the extracted examples are exhausted. This generator tests whether the task recognition and replay mechanisms are able to gradually adapt to changing distributions.

### E.2 DDM vs KSWIN

Table 8 compares the average classification accuracy of TSR with DDM and KSWIN as drift detectors over benchmark image datasets. When KSWIN is used, the classification accuracy is higher on average than DDM. When DDM is used, there is a very high variance, suggesting that DDM relies heavily on the ordering of tasks or examples to reliably detect concept drift in a stream of logits. We therefore use KSWIN for all other experiments.

Table 8: Comparison of mean classification accuracy over benchmark image datasets when DDM and KSWIN are used as drift detectors for TSR.

| Drift Detector | MNIST | Fashion MNIST | CIFAR-100 | COIL-100 |
| --- | --- | --- | --- | --- |
| DDM | 88.07 (3.02) | 73.45 (12.96) | 6.47 (11.18) | 8.11 (10.81) |
| KSWIN | 88.23 (2.43) | 73.54 (2.96) | 6.89 (1.05) | 8.23 (1.11) |

### E.3 TSR Sensitivity

In this section, we perform sensitivity tests on TSR by varying two TSR hyperparameters, $\mathcal{R}_{max}$ and $n$. Replay per batch, denoted $\mathcal{R}_{max}$, is a hyperparameter in TSR defining the number of replayed examples that should be introduced to the next batch if drift is detected on a task. For instance, if a batch of size $B$ is the next training batch and drift on a previous task is detected, the new batch size is $B + \mathcal{R}_{max}$. Table 9 shows the average classification accuracy of TSR on benchmark datasets with varied $R_B$. Table 10 shows the average training time for TSR with varied $\mathcal{R}_{max}$.

Table 9: Average accuracy for TSR with varied $\mathcal{R}_{max}$ on a range of benchmark image datasets.

| Dataset | 10 | 20 | 30 | 40 |
| --- | --- | --- | --- | --- |
| MNIST | 88.01 (2.23) | 88.23 (2.43) | 88.34 (2.13) | 88.52 (2.71) |
| Fashion MNIST | 73.05 (2.91) | 73.54 (2.96) | 73.99 (2.82) | 74.03 (2.63) |
| CIFAR-100 | 6.86 (1.12) | 6.89 (1.05) | 6.84 (1.16) | 6.82 (1.25) |
| COIL-100 | 8.22 (1.01) | 8.23 (1.11) | 8.25 (1.15) | 8.18 (1.13) |

Table 10: Average training time in seconds for TSR with varied $\mathcal{R}_{max}$ on a range of benchmark image datasets.

| Dataset | 10 | 20 | 30 | 40 |
|---|---|---|---|---|
| MNIST | 5.41 (1.98) | 6.12 (1.57) | 6.67 (1.31) | 6.89 (1.63) |
| Fashion MNIST | 6.55 (1.55) | 6.74 (1.23) | 7.14 (1.18) | 7.53 (1.17) |
| CIFAR-100 | 40.76 (5.42) | 43.49 (5.58) | 48.12 (5.45) | 50.98 (5.71) |
| COIL-100 | 29.34 (6.36) | 30.72 (6.09) | 31.95 (6.53) | 33.01 (6.48) |

Table 11 shows the average classification accuracy of TSR as it changes with varied validation batch size $n$. There is not a statistically significant difference in mean accuracy between $n = 50$ and $n = 200$ over any dataset. Table 12 shows the mean training time for TSR for varied $n$.

Table 11: Average accuracy for TSR with varied $n$ on a range of benchmark image datasets.

| Dataset | 50 | 100 | 150 | 200 |
|---|---|---|---|---|
| MNIST | 88.23 (2.43) | 88.52 (2.79) | 88.64 (2.59) | 88.62 (2.48) |
| Fashion MNIST | 73.54 (2.96) | 73.08 (2.25) | 73.06 (2.51) | 73.89 (2.43) |
| CIFAR-100 | 6.89 (1.05) | 6.81 (1.55) | 6.99 (1.23) | 7.01 (1.71) |
| COIL-100 | 8.23 (1.11) | 8.29 (1.67) | 8.32 (1.72) | 8.33 (1.62) |

### E.4 TF-TSR SENSITIVITY

In this section, we present the sensitivity of TF-TSR to its hyperparameters in terms of accuracy and the number of recognized tasks. First, Table 13 shows the number of recognised tasks as task creation sensitivity $\delta$ is varied between 0.2 and 0.8 on benchmark image datasets. We observe that the number of tasks increases with $\delta$.

Second, we conduct sensitivity tests for two key hyperparameters in TF-TSR: the maximum size of the validation set $\mathcal{V}_{max}$, and the task creation sensitivity $\delta$. By varying these hyperparameters and observing the corresponding effects on the model's performance, we aim to investigate their influence and identify the optimal configurations. Figure 7 provides the accuracy of TF-TSR on synthetic datasets for a range of $p$ and $q$ when $\mathcal{V}_{max}$ and $\delta$ are varied.

Table 12: Average training time in seconds for TSR with varied $n$ on a range of benchmark image datasets.

| Dataset | 50 | 100 | 150 | 200 |
|---|---|---|---|---|
| MNIST | 6.12 (1.57) | 6.64 (1.62) | 6.77 (1.36) | 6.89 (1.62) |
| Fashion MNIST | 6.74 (1.23) | 7.22 (1.39) | 7.44 (1.23) | 7.53 (1.22) |
| CIFAR-100 | 43.49 (5.58) | 50.64 (5.44) | 55.12 (5.54) | 60.98 (5.21) |
| COIL-100 | 30.72 (6.09) | 42.08 (6.87) | 48.45 (6.21) | 55.01 (6.37) |

Table 13: Average number of recognized tasks for TF-TSR with varied $\delta$.

| $\delta$ | 0.2 | 0.4 | 0.6 | 0.8 |
|---|---|---|---|---|
| MNIST | 3.61±1.31 | 4.16±1.38 | 4.88±1.27 | 5.35±1.36 |
| Fashion MNIST | 4.67±1.36 | 5.41±1.44 | 5.84±1.50 | 6.16±1.17 |
| CIFAR-100 | 6.16±3.95 | 11.14±7.04 | 17.16±8.14 | 19.39±7.25 |
| COIL-100 | 14.24±9.19 | 24.27±8.26 | 29.20±9.26 | 37.84±9.92 |
| Mini ImageNet | 11.73±6.60 | 20.49±7.75 | 25.73±8.69 | 32.08±10.27 |

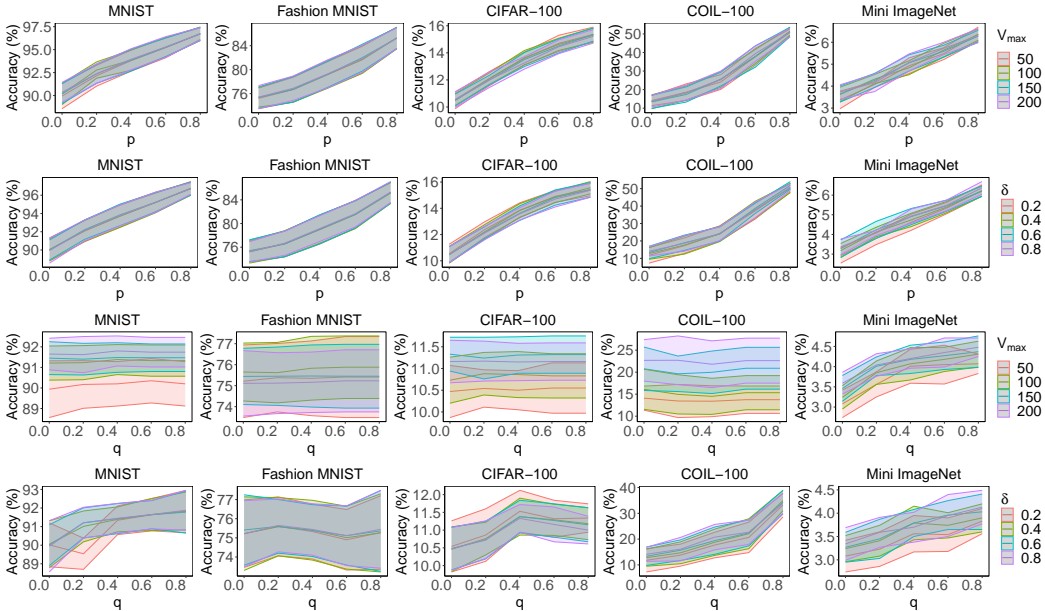

Figure 7: Accuracy comparison of TF-TSR on synthetic image datasets with $p$ ranging from $0.0$ to $0.8$ with varied $\mathcal{B}_{max}$, $\mathcal{V}_{max}$, $\mathcal{R}_{max}$, and $\delta$. Each row varies a single hyperparameter.

