# OpenReview forum: "Time-Sensitive Replay for Continual Learning"
_ICLR.cc/2024/Conference — Submitted to ICLR 2024_

### Official Review · Reviewer_AYp8 · 2023-10-28

**Soundness:** 2 fair
**Presentation:** 1 poor
**Contribution:** 2 fair
**Rating:** 3
**Confidence:** 4

**Summary:**

The paper presents continuous learning methods in settings where the task corresponding to each sample is known and in settings where it is not. The authors perform extensive numerical experiments that show the superiority of the method presented.

**Strengths:**

The authors provide a good review of replay-based continuous learning methods.

Methods are proposed for two settings where the task index is known and where the task index is not known.

The authors present multiple numerical experiments.

**Weaknesses:**

The authors review the literature of continual learning based on replay, but does not explain about other methods that follow other strategies such as regularization-based methods. This review of methods shows the contributions of the proposed method versus replay methods but not versus other strategies.

In the introduction, the authors refer to equations that appear later in the text, and includes notation that makes the introduction difficult to understand.

The reviewer believes that the paragraph on confidence in Section 3 could be better explained. For example, the symbol $n$ appears in the first line and is explained in line 4 which makes it difficult to understand.

The quality of the plots can be improved. The reviewer thinks that it is not clear what the authors want to show. The colors are not well distinguished and the plots are too small.

**Questions:**

In the third paragraph of the introduction, the authors introduce the term "evolving" in the sentence "This enables the method to adapt to evolving data streams without prior knowledge of future tasks." What does evolving mean? Does it refer to any assumptions about the distribution of tasks over time?

According to the definition of task given in the preliminaries, could task be any dataset? Is there no assumption between consecutive tasks, or about the distribution between tasks, or that tasks are related?

The reviewer believes that the paragraph on confidence in Section 3 could be better explained. For example, the symbol n appears in the first line and is explained in line 4 which makes it difficult to understand.

The reviewer thinks there is a typo in the equation of a in section 3, I think it would be $l_j$ instead of $l_i$

Is $n$ in the second paragraph of section 4.1 different from the $n$ in section 3?

At the end of the second paragraph of section 4.1 it is explained that "the oldest logit is discarded". Why the oldest? Is there time dependence in the distribution?

---

> ### Author Response · Authors · 2023-11-22
> **Response to Reviewer AYp8**
>
> We thank the reviewer for their time and thorough analysis of our paper. We are happy that the reviewer finds our experiments satisfactory and our review of replay models suitable. We address the reviewer’s questions as follows:
>
> > In the third paragraph of the introduction, the authors introduce the term "evolving" in the sentence "This enables the method to adapt to evolving data streams without prior knowledge of future tasks." What does evolving mean? Does it refer to any assumptions about the distribution of tasks over time?
>
> By “evolving” we refer to the changing distributions of examples over a data stream, meaning that we expect the frequency of examples belonging to different classes to change. We have clarified this in our revisions.
>
> > According to the definition of task given in the preliminaries, could task be any dataset? Is there no assumption between consecutive tasks, or about the distribution between tasks, or that tasks are related?
>
> Because of the nature of our experiments and our method, we assume that a dataset contains examples of a constant shape with the same modality. A task is instead defined as a distribution of classes. We clarify this in our revision.
>
> > The reviewer believes that the paragraph on confidence in Section 3 could be better explained. For example, the symbol n appears in the first line and is explained in line 4 which makes it difficult to understand.
>
> We appreciate that this may cause some confusion and have corrected this in our revision.
>
> > The reviewer thinks there is a typo in the equation of a in section 3.
>
> We thank the reviewer for pointing out this typo. It has been corrected in our revision.
>
> > Is n in the second paragraph of section 4.1 different from the n in section 3?
>
> No, n refers to the number of validation examples tested to obtain a sequence of logits of length n in both instances. We clarify this in our revision.
>
> > At the end of the second paragraph of section 4.1 it is explained that "the oldest logit is discarded". Why the oldest? Is there time dependence in the distribution?
>
> The oldest logit is discarded because it has the least relevance to monitoring currently occurring prediction drifts. We are more interested in recent changes than changes that have already occurred. We address this in our revision.
>
> Additional changes have been highlighted in the revised paper.

---

> > ### Comment · Reviewer_AYp8 · 2023-11-22
> >
> > Thank you for the responses to the comments. The authors have answered all the questions. However, I will maintain my score.

---

### Official Review · Reviewer_qppC · 2023-10-30

**Soundness:** 3 good
**Presentation:** 3 good
**Contribution:** 2 fair
**Rating:** 5
**Confidence:** 4

**Summary:**

The paper develop a new replay-based approach, called Time-Sensitive Replay (TSR), to tackle catastrophic forgetting in continual learning settings. The core idea is to minimize redundancy by reintroducing previous examples when the model's predictions significantly deviate from past tasks while learning new tasks. As a result, the number of stored samples can be reduced without compromising the model performance. The idea is also applicable to a task-free scenario, Task-Free TSR (TF-TSR). Experimental results on benchmark datasets are provided to demonstrate that TSR is particularly effective in improving the learning speed while maintaining high-performance accuracies.

**Strengths:**

1. The paper can be followed straightforwardly.

2. Experiments demonstrate that the idea is effective.

**Weaknesses:**

1. The setting is somewhat contrived and it is not clear how practical it is and its relevance to real-world applications.

2. Comparison against prior methods is limited which makes judgment about the competence of the method challenging.

3. The code is not provided which makes judgment about reproducibility challenging.

**Questions:**

1. Experiments are performed in synthetic settings using benchmarks that are common but do not have a natural notion of time. Given the proposed setting, why haven't experiments been performed on suitable real-world datasets that have a notion of time?

2. Continual learning has a rich literature and there are many works on the topic. What is the justification for selecting the methods that have been used in the paper? Do they have SOTA performance? It is very common in CL literature to provide an extensive comparison.



============Post Rebuttal=============

Thank you for preparing the rebuttal. I maintain my initial rating.

---

> ### Author Response · Authors · 2023-11-22
> **Response to Reviewer qppC**
>
> We thank the reviewer for their time and thorough analysis of our paper. We are happy that the reviewer finds our paper straightforward and our experiments effective. We address the reviewer’s questions as follows:
>
> > Experiments are performed in synthetic settings using benchmarks that are common but do not have a natural notion of time. Given the proposed setting, why haven't experiments been performed on suitable real-world datasets that have a notion of time?
>
> We wanted to remain consistent with theoretical CL work, many of which compare with image datasets constructed to be introduced over time. It would be interesting future work to extend our method for datasets that have time information and where time is important, but doing so would likely require a recurrent architecture that in our opinion falls beyond the scope of this work.
>
> > Continual learning has a rich literature and there are many works on the topic. What is the justification for selecting the methods that have been used in the paper? Do they have SOTA performance? It is very common in CL literature to provide an extensive comparison.
>
> We selected models with state-of-the-art performance for both replay and regularisation. However, we appreciate and agree with the comment that TSR and TF-TSR should be compared with more recent methods, and have provided additional experiments in the revised paper. We provide three other continual learning methods: Experience Replay (ER), Continual Learning by Modelling Intra-Class Variation (MOCA) and Proxy-based Contrastive Replay (PCR).
>
> > The code is not provided which makes judgment about reproducibility challenging.
>
> A link to an anonymous repository containing our code is provided on Page 2 in the final paragraph of the Introduction and in Section E.1 on Page 15.
>
> Additional changes have been highlighted in the revised paper.

---

### Official Review · Reviewer_D8wn · 2023-10-30

**Soundness:** 3 good
**Presentation:** 2 fair
**Contribution:** 2 fair
**Rating:** 3
**Confidence:** 3

**Summary:**

This paper investigates replay-based continual learning and seeks to reduce the number of replayed examples during training while maintaining accuracy. To this end, this paper proposes a replay method named Time-Sensitive Replay (TSR), which first detects drifts in the model prediction when learning a new task and then introduces the replay data for training to mitigate forgetting when drifts are detected. A variant called TF-TSR is further proposed to handle the task-free setting where the task boundary is unknown. Experiments are conducted to show that the proposed algorithms can achieve comparable performance with baseline methods but in a shorter training time.

**Strengths:**

1. The idea of managing data replay at a smaller time scale, i.e., per training data batch, and investigating when to replay is interesting.

2. A variant of the algorithm is also proposed to handle the task-free setting.

3. The experimental results show that the number of replayed examples can be successfully reduced.

**Weaknesses:**

1. While I appreciate the authors' effort in investigating the problem about when to replay, one of my main concerns is about the availability and complexity of using the forgetting performance of all previous tasks as the feedback signal to determine if replay is needed for the previous tasks. To detect the distribution shifts, evaluations of the current model on the validation datasets for all previous tasks are required at each time step (or each batch) to determine if forgetting happens for any previous task.

- First, whether such a validation dataset is available for each task is questionable. Without this, the forgetting performance cannot be evaluated.
- Second, even if we have such validation datasets, evaluating the performance on all previous tasks using the validation data at each time step will introduce a lot of computational complexity, especially when the number of tasks is large. The detection complexity can be much higher than the replay complexity. Even though an analysis of the theoretical complexity is provided in the appendix, it is unclear if the proposed algorithm can achieve better computational complexity than other replay-based methods in theory.

2. Another main concern is about the empirical evaluation.
- First, in the comparison, even if the proposed method has a shorter training duration, the final accuracy can be, for example, 2\% lower than CEC on MNIST. This somehow indicates that the condition to determine when to replay is not so helpful. In other words, the proposed algorithm does not tell us correctly when to replay, and more data replay clearly improves the performance as in CEC.
- Second, only two baselines (among which only one is replay-based) are considered in the paper, which are also relatively outdated. Considering that there are many advanced replayed-based CL methods out there, more baseline methods should be investigated here. Specifically, the question here is that if we can select the replay samples wisely, do we still need to care about when to replay? Maybe the answer is still yes, but this investigation will further justify the contribution of this paper.

3. The writing can be improved and there are multiple typos, e.g., $l_i^T$ should be $l_j^T$ in calculating the moving average, $l_k^{T_1} $ should be $l_k^{T_i}$ in "Model Update and Validation".

**Questions:**

Please see the weaknesses above


####################**Post Rebuttal**####################

Thanks for the authors' rebuttal. However, my concerns on the validation data remain and the performance comparison with newly added methods seems not satisfying. I will maintain my original rating.

---

> ### Author Response · Authors · 2023-11-22
> **Response to Reviewer D8wn**
>
> We thank the reviewer for their time and thorough analysis of our paper. We are happy that the reviewer finds our idea interesting and that a task-free implementation is useful. We address the reviewer’s questions as follows:
>
> >  First, whether such a validation dataset is available for each task is questionable. Without this, the forgetting performance cannot be evaluated.
>
> We agree that a pre-established validation buffer severely impacts the practicality of the TSR method. Similarly to TSR’s reliance on a knowledge of task information, in most real-world cases, it is unrealistic to expect a priori samples of a data stream. Both of these limitations are addressed in the TF-TSR method. Page 5 of our paper contains a paragraph on the TF-TSR Validation and Replay Buffer changes. Here, we acknowledge that a validation set cannot be expected prior to training, so validation examples must be stored in memory as new data arrives. In this instance, there is no additional information or data required by TF-TSR prior to training. We appreciate that this aspect should be highlighted more in the paper, and so have made revisions on Page 5. We provide further experiments varying the size of the validation buffer in the Appendix on Page 18.
>
> > Second, even if we have such validation datasets, evaluating the performance on all previous tasks using the validation data at each time step will introduce a lot of computational complexity, especially when the number of tasks is large. The detection complexity can be much higher than the replay complexity. Even though an analysis of the theoretical complexity is provided in the appendix, it is unclear if the proposed algorithm can achieve better computational complexity than other replay-based methods in theory.
>
> Evaluations are required at each batch to monitor for prediction drift over previous tasks. On Pages 13 and 14 of the Appendix, we discuss the theoretical complexity of this process. We appreciate that it is unclear how the replay must be reduced to provide benefit despite consistent validation tests, and so have made revisions in the paper. Furthermore, we have provided additional discussion in this section about the complexity of forward-pass validation tests. We describe the process of sampling from the validation buffer to reduce this complexity further. Moreover, on Page 9, we empirically compare the training time of TSR and TF-TSR with existing baselines and find that despite the expense required to monitor performance over previous tasks, our methods nonetheless achieve a lesser training time.
>
> >First, in the comparison, even if the proposed method has a shorter training duration, the final accuracy can be, for example, 2% lower than CEC on MNIST. This somehow indicates that the condition to determine when to replay is not so helpful. In other words, the proposed algorithm does not tell us correctly when to replay, and more data replay clearly improves the performance as in CEC.
>
> Even if the mean accuracy may sometimes be lower for our experiments than other full replay models, our results always fall within the error of baseline evaluations and there is never a statistically significant difference in our results. We have made some revisions in our experiments section to emphasise this.
>
> >Second, only two baselines (among which only one is replay-based) are considered in the paper, which are also relatively outdated. Considering that there are many advanced replayed-based CL methods out there, more baseline methods should be investigated here. Specifically, the question here is that if we can select the replay samples wisely, do we still need to care about when to replay? Maybe the answer is still yes, but this investigation will further justify the contribution of this paper.
>
> We appreciate and agree with the comment that TSR and TF-TSR should be compared with more recent methods. We provide three other continual learning methods: Experience Replay (ER), Continual Learning by Modelling Intra-Class Variation (MOCA) and Proxy-based Contrastive Replay (PCR). These additional results demonstrate that TF-TSR achieves a classification accuracy that is not significantly different to current state-of-the-art methods with lower time requirements. An opportunity for future work may be to investigate whether sampling algorithms may enhance TSR.
>
> > The writing can be improved and there are multiple typos.
>
> We agree with the reviewer that there are a couple of oversights in our writing. We have corrected these oversights in the revised paper.
>
> Additional changes have been highlighted in the revised paper.

---

### Official Review · Reviewer_Xzud · 2023-11-05

**Soundness:** 2 fair
**Presentation:** 2 fair
**Contribution:** 2 fair
**Rating:** 3
**Confidence:** 4

**Summary:**

This paper proposes an approach called Time-Sensitive Replay (TSR), which aims reduce the computational load of continual learning while maintaining learning performance. TSR replays data from a particular task only when it detects that this task has started being forgotten. The forgetting detection takes place with the help of stored validation data from each task.

**Strengths:**

- I found the main idea of the paper (reducing computations while maintaining accuracy) to be interesting and worth examining.

**Weaknesses:**

- A considerable limitation of the paper is that it ignores cases where the data distribution changes continuously over time. In such cases, task boundaries cannot be defined or detected. In my opinion, most real-world applications of continual learning do not have task boundaries, since the distribution is constantly changing (e.g., weather, climate, stock market, hospital occupancy, etc.).
- The number of baselines used in the evaluation is insufficient, in my opinion (especially for a major venue like ICLR).
- I am not sure if the comparison of TSR and TF-TSR with other baselines is fair. TSR requires the storage of validation data, while other methods do not necessarily do so, and I could not find any part of the text that explains whether this discrepancy was taken into account in the evaluation.
- The references for MNIST and FashionMNIST are incorrect I think. (This point has not affected my evaluation of the paper.)

**Questions:**

- Can you explain if and how you took into account the size of the validation memory in the evaluation?
- Why did you not compare with other very simple and efficient baselines (e.g., simple experience replay)?

---

> ### Author Response · Authors · 2023-11-22
> **Response to Reviewer Xzud**
>
> We thank the reviewer for their time and thorough analysis of our paper. We are happy that the reviewer finds the premise and motivation of the TSR method strong. We particularly appreciate the reviewer’s suggestion to compare with further continual learning baselines. Additional experiments are provided to address this. We address the reviewer’s questions as follows:
>
> > Can you explain if and how you took into account the size of the validation memory in the evaluation?
>
> We agree that a pre-established validation buffer severely impacts the practicality of the TSR method. Similarly to TSR’s reliance on a knowledge of task information, in most real-world cases, it is unrealistic to expect a priori samples of a data stream. Both of these limitations are addressed in the TF-TSR method. Page 5 of our paper contains a paragraph on the TF-TSR Validation and Replay Buffer changes. Here, we acknowledge that a validation set cannot be expected prior to training, so validation examples must be stored in memory as new data arrives. In this instance, there is no additional information or data required by TF-TSR prior to training. We appreciate that this aspect should be highlighted more in the paper, and so we have made adjustments in this section. We provide further experiments varying the size of the replay buffer in the Appendix on Page 18.
>
> > Why did you not compare with other very simple and efficient baselines (e.g., simple experience replay)?
>
> We thank the reviewer for pointing out this oversight. Our perspective was that comparing with more recent and superior replay methods was sufficient. We have provided additional experiments to remedy this. In addition to our current baselines, we provide three other continual learning methods: Experience Replay (ER), Continual Learning by Modelling Intra-Class Variation (MOCA) and Proxy-based Contrastive Replay (PCR). These additional results demonstrate that TF-TSR achieves a classification accuracy that is not significantly different to current state-of-the-art methods with lower time requirements.
>
> > A considerable limitation of the paper is that it ignores cases where the data distribution changes continuously over time. In such cases, task boundaries cannot be defined or detected. In my opinion, most real-world applications of continual learning do not have task boundaries, since the distribution is constantly changing.
>
> We agree that it is important to consider changing distributions in a data stream for continual learning settings. We intended that distribution changes are represented as tasks in our task-sequential setting, where examples may abruptly shift throughout the data stream. We also employed synthetic datasets to adjust the distributions and how distributions change over each benchmark image dataset. These synthetic datasets are described on Page 6 and used on Pages 8 and 9, where we observe that TF-TSR can successfully detect tasks even under these more challenging settings. We appreciate that this should be made clear in the paper, and so have made revisions to the definition of a task on Page 3 and have added a further description of synthetic datasets in Page 18 in the Appendix.
>
> Additional changes have been highlighted in the revised paper.

---

> > ### Comment · Reviewer_Xzud · 2023-11-22
> > **Rebuttal Acknowledgement**
> >
> > Thank you for your response, and I appreciate your attempts to improve the manuscript.

---

### Meta-Review · Area_Chair_QeCo · 2023-12-05

**Metareview:**

(a) Summarize the scientific claims and findings of the paper based on your own reading and characterizations from the reviewers.
- The authors propose a novel replay method for continual learning. Instead of "blindly" replaying all examples (from the buffer), the method only replays examples from tasks currently being forgotten. The authors also propose a task-free approach by exploiting a task inference idea.

(b) What are the strengths of the paper?
- The paper is well motivated (one of CL's most interesting aspects is reducing computation).
- Proposing a task-free method has potential practical benefits

(c) What are the weaknesses of the paper? What might be missing in the submission?
- One of the two proposed approaches (TSR) requires extra computations to detect tasks that are being forgotten. Overall, it is unclear if the proposed approach saves computation compared to other (vanilla) replay methods.
- Originally, baseline methods were missing in the paper although that has been partially fixed in the latest version of the paper.
- There are other potential limitations of the approach, including assuming the existence of tasks (as opposed to, e.g., continuous data drift), but in all fairness, this is the setting most CL works under.

**Justification For Why Not Higher Score:**

I find that this paper has most elements of a good paper, but its presentation still requires significant modifications.

Note that I didn't find the review of reviewer AYp8 to be too informative and down weighted it.

**Justification For Why Not Lower Score:**

N/A

---

### Decision · Program_Chairs · 2024-01-16

Reject